# Functionally Regionalized Knowledge Transfer for Low-resource Drug Discovery

**Huaxiu Yao**[1]*, **Ying Wei**[2], **Long-Kai Huang**[3], **Ding Xue**[3]
**Junzhou Huang**[4], **Zhenhui Li**[5]
[1]Stanford University, [2]City University of Hong Kong, [3] Tencent AI Lab
[4]University of Texas at Arlington, [5]Pennsylvania State University
[1]huaxiu@cs.stanford.edu, [2]yingwei@cityu.edu.hk, [3]hlongkai@gmail.com, dingxue@tencent.com
[4]jzhuang@uta.edu, [5]jessieli@ist.psu.edu

## Abstract

More recently, there has been a surge of interest in employing machine learning approaches to expedite the drug discovery process where virtual screening for hit discovery and ADMET prediction for lead optimization play essential roles. One of the main obstacles to the wide success of machine learning approaches in these two tasks is that the number of compounds labeled with activities or ADMET properties is too small to build an effective predictive model. This paper seeks to remedy the problem by transferring the knowledge from previous assays, namely in-vivo experiments, by different laboratories and against various target proteins. To accommodate these wildly different assays and capture the similarity between assays, we propose a functional rationalized meta-learning algorithm *FRML* for such knowledge transfer. FRML constructs the predictive model with layers of neural sub-networks or so-called functional regions. Building on this, FRML shares an initialization for the weights of the predictive model across all assays, while customizes it to each assay with a region localization network choosing the pertinent regions. The compositionality of the model improves the capacity of generalization to various and even out-of-distribution tasks. Empirical results on both virtual screening and ADMET prediction validate the superiority of FRML over state-of-the-art baselines powered with interpretability in assay relationship.

## 1 Introduction

Drug discovery brings new candidate medications to billions of people, helping them live longer, healthier and more productive lives. One crux step in drug discovery is virtual screening, which is a fast and cost-effective method that computationally predicts the activity value of a compound against the target protein of a disease. As shown in Figure 1(a), the hits screened out of large drug libraries of compounds by a virtual screening algorithm are further empirically validated against their in-vivo activities, resulting in leads. After optimizing the ADMET properties (absorption, distribution, metabolism, excretion and toxicities) of the leads, we obtain the drug candidates.

There have been both traditional machine learning [33] and deep learning approaches [4] devoted to virtual screening, while the prediction performance (i.e., the hit rate) is far from satisfactory. The crucial challenge lies in that the number of training compounds whose activities have been tested against the target protein of focus is severely limited. Though state-of-the-art deep learning algorithms typically rely on supervision in the form of thousands to millions of annotated data, it is highly expensive and almost impossible for in-vivo experiments to collect a sufficient set of drug

---

*Part of the work was done when H.Y. was a student at Penn State University, correspondence to: Y.W.

35th Conference on Neural Information Processing Systems (NeurIPS 2021).

compounds with activity labels. In fact, virtual screening as a computational pre-screening method is desired precisely because of the prohibitive costs of an in-vivo experiment (i.e., an assay). Fortunately, previous assays conducted by different laboratories around the world towards a wide variety of diseases with different biological target proteins together provide a rich repository for learning the interactions between a protein and a compound. For example, as COVID-19 and SARS share high amino acid sequence identity, previous assays against SARS 3CLpro and PLpro proteases contribute a lot to learning a predictive model for COVID-19 [16].

We are highly motivated to transfer the knowledge of interactions from this repository to address the scarcity of labeled compounds in the assay against the target protein of our focus, which we name as the target assay for convenience. The challenges of such knowledge transfer are two-fold: (1) how to share the transferable knowledge but meanwhile accommodate the wide variance between assays, and (2) how to adequately identify the nearest neighbor assays to the target assay to reduce the risk of negative transfer. Since assays are from different institutions and against various target proteins,

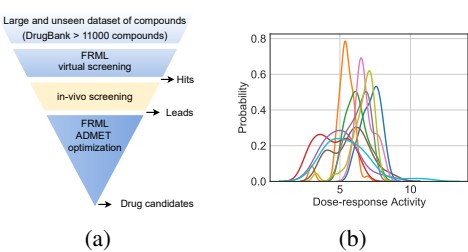

(a)                  (b)

Figure 1: (a): Workflow for discovery of drug candidates using virtual screening. (b): Distributions of activity values for 10 randomly selected assays.

the compounds tested and the distribution of activity values vary a lot from assay to assay. As evidenced in Figure 1(b), there exists a large discrepancy between distributions of activity values for 10 randomly selected assays. The prevalent fine-tuning strategy in transfer learning [21], trains a single model on previous assays and fine-tunes it to the target assay – it struggles in predicting accurately for each assay and confuses the most similar assays to the target with the others.

Gradient-based meta-learning [7] has been a promising practice, which learns from previous assays an initialization for a shared predictive model and adapts the model from this initialization to each assay. while the initialization is learned so that the adapted model of each assay generalizes well on testing compounds, maintaining a shared initialization is still insufficient to handle wildly varying assays [37] and pinpoint the most similar assays. Recent efforts on heterogeneous meta-learning deal with this issue by modulating the shared initialization to different assays via task embedding [20, 35, 37]. Instead of only differentiating initializations, motivated by compositionality and brain functional specialization in neuroscience [5, 26], we aim to push ahead with distinguishing neural sub-networks, or so-called *functional regions*, each of which consists of a disparate set of parameters. This advancement brings at least the following two benefits. First, the similarity between assays is more accurately measured in a divide-and-conquer manner – only modulation for the initialization weights in those overlapping regions between two assays are considered for comparison. Second, the reduced parameter space prevents the predictive model from overfitting to a limited set of training compounds.

We name the resulting meta-learning algorithm as FRML. The predictive model of the FRML is dissected into a sequence of hierarchically organized functional regions. Provided with an assay, the contrastive assay representation network forwards the learned assay embedding to a region localization network. The region localization network locates the most relevant functional regions for the assay in a recurrent manner, to be consistent with the hierarchical organization of functional regions. In the stage of meta-training on previous assays, both the region localization network and the weights for initializations of all functional regions are jointly learned. When it comes to meta-testing, FRML quickly adapts to the target assay via easy assembly of the located regions.

We summarize our major contributions as follows. (1) We propose a novel meta-learning algorithm FRML, which pushes a step forward from differentiating initializations to differentiating neural sub-networks between tasks; (2) We demonstrate the effectiveness of FRML on not only virtual screening but also the task of ADMET prediction. (3) FRML respects the key principle of machine learning models in healthcare – it is interpretable in the relationship between assays.

## 2 Notations and Problem Definition

In this section, we define some notations and discuss our problem. In drug prediction, we consider each task $\mathcal{T}_i$ as an assay which refers to an in-vivo experiment on a group of compounds, and all tasks

are sampled from the distribution $p(\mathcal{T})$. Note that we use either task or assay alternatingly in the remainder of this paper. Assuming that we have $N$ historical assays $\{\mathcal{T}_i\}_{i=1}^N$ as meta-training assays, we aim to generalize a meta-learner from these meta-training assays and quickly adapt it to unseen target assays $\{\mathcal{T}_t\}_{t=1}^{N_t}$ even with limited amount of annotated data. Here, we define the process of learning well-generalized meta-knowledge from the meta-training assays as the *meta-training phase* and the adaption process on the target assays as the *meta-testing phase*.

Concretely, for each task $\mathcal{T}_i$, a support set of training samples $\mathcal{D}_i^s = \{\mathbf{X}_i^s, \mathbf{Y}_i^s\} = \{(\mathbf{x}^s, \mathbf{y}^s)_{i,j}\}_{j=1}^{n_i^s}$ and a query set of testing samples $\mathcal{D}_i^q = \{\mathbf{X}_i^q, \mathbf{Y}_i^q\} = \{(\mathbf{x}^q, \mathbf{y}^q)_{i,j}\}_{j=1}^{n_i^q}$ are sampled from $\mathcal{T}_i$, where $n_i^s$ and $n_i^q$ represent the number of support and query samples, respectively. Denote that the feature space is $\mathcal{X}$ and the label space is $\mathcal{Y}$, a predictive model (a.k.a., base learner) $f : \mathbf{x} \mapsto \hat{\mathbf{y}}$ is defined to map a sample $\mathbf{x} \in \mathcal{X}$ to its predicted value $\hat{\mathbf{y}} \in \mathcal{Y}$. For each task $\mathcal{T}_i$, the base learner $f$ is updated from the initialization $\theta_0$ by minimizing the expected empirical loss $\mathcal{L}$ on $\mathcal{D}_i^s$, i.e., $\min_\theta \mathcal{L}(\theta; \mathcal{D}_i^s)$, resulting in the optimal parameters $\theta_i$. Specifically, the loss function $\mathcal{L}$ is defined as mean square error (i.e., $\sum_{(\mathbf{x},\mathbf{y})\in\mathcal{D}_i^s} \|f_\theta(\mathbf{x})-\mathbf{y}\|_2^2$) or cross-entropy loss (i.e., $-\sum_{(\mathbf{x},\mathbf{y})\in\mathcal{D}_i^s} \log p(\mathbf{y}|\mathbf{x}, f_\theta)$) for regression and classification problems, respectively. In the meta-training phase, the query sets $\{\mathcal{D}_i^q\}_{i=1}^N$ of all meta-training assays are used to optimize the initialization of the base learner, so that the final initialization $\theta_0^*$ is well-generalized. $\theta_0^*$ can be further adapted to each meta-testing task $\mathcal{T}_t$ via the corresponding support set $\mathcal{D}_t^s$. Formally, we define our problem as,

$$\hat{\mathbf{Y}}_t^q = \arg\max_{\mathbf{Y}_t^q} p(\mathbf{Y}_t^q | \mathbf{X}_i^q, \mathcal{D}_t^s, f_{\theta_0^*}). \tag{1}$$

The well-generalized model initial weights $\theta_0^*$ encrypt the comprehensive knowledge learned from meta-training assays. We will detail how to learn $\theta_0^*$ in Section 3.

## 3 Methodology

In this section, we introduce the proposed framework FRML whose overview is illustrated in Figure 2. The goal of FRML is to improve the generalization ability for a wide range of and even out-of-distribution target assays with limited training samples via discriminating functional regions between assays. To achieve this goal, we dissect the base learner into a sequence of functional regions. Given a new assay, we propose a region localization network taking the learned assay representation as input to locate and assemble the most relevant functional regions. Subsequently, FRML can be quickly adapted to the novel assay on the assembled functional region set. In the following subsections, we will first discuss the predictive models for virtual screening and ADMET classification as the base learner and our meta-learning pipeline. Then we elaborate the details of three key components (i.e., assay representation learning, localization strategy, and region localization network).

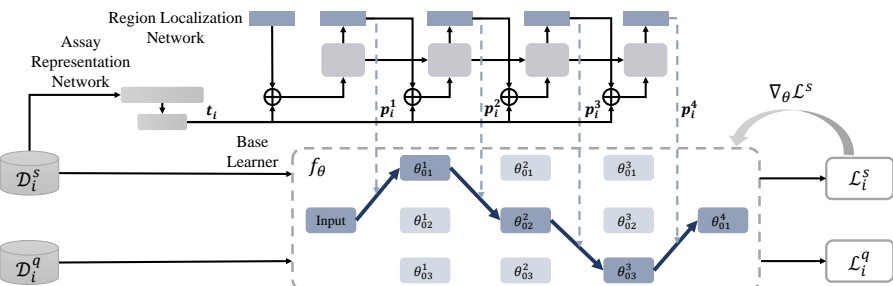

Figure 2: Overview of the proposed FRML. In each assay $\mathcal{T}_i$, the recurrent region localization network, guided by its learned representation $\mathbf{t}_i$, locates the most relevant functional regions (darker blocks) and assembles them (trace: input$\rightarrow \theta_{01}^1 \rightarrow \theta_{02}^2 \rightarrow \theta_{03}^3 \rightarrow \theta_{01}^4$) in the dissected base learner $f_\theta$.

### 3.1 Predictive Models for Drug Discovery and Gradient-based Meta-Learning

We build predictive models for virtual screening and ADMET prediction, both of which are crucial for drug discovery. The input to the predictive models is a drug compound represented by 1024 dimensional Morgan fingerprints [28], i.e., $\mathbf{x} \in \mathbb{R}^{1024}$. For virtual screening, the output is the activity value of the compound against the target protein in this assay, i.e., $y \in \mathbb{R}$, while the output for

ADMET prediction could be a discrete category or a real value. In our empirical study, we only consider those ADMET prediction tasks of classification, i.e., $y \in \mathcal{C}$, where $\mathcal{C}$ denotes the set of property categories. Building on these, we construct a neural network consisting of two fully connected layers as the predictive model, which also serves as the base learner $f$. We denote the weights for the base learner $f$ to be $\theta$.

With the base learner $f$, we introduce gradient-based meta-learning as the backbone meta-learning framework, which regards the initialization $\theta_0$ for the base learner as the transferable knowledge. Apparently, it enjoys the advantage of being independent of problem types. Specifically, here we illustrate the gradient-based meta-learning by using model-agnostic meta-learning (MAML) [6] as an example. In the meta-training phase, MAML obtains the assay-specific model for each assay $\mathcal{T}_i$ by updating the parameters $\theta$ via the support set $\mathcal{D}_i^s$ in a few gradient steps starting from $\theta_0$, i.e.,

$$\theta_i = \theta_0 - \alpha \nabla_\theta \mathcal{L}(\theta; \mathcal{D}_i^s). \tag{2}$$

Here $\alpha$ denotes the learning rate for assay adaptation. Though only one gradient step is presented as exemplary in Eqn. (2), it is easy to extend to several gradient steps. The crux is to evaluate the adapted assay-specific model $\theta_i$ on the query set $\mathcal{D}_i^q$ and leverage the result as a feedback to meta-update the initializations $\theta_0$ as,

$$\theta_0 \leftarrow \theta_0 - \beta \frac{1}{N} \sum_{\mathcal{T}_i \in p(\mathcal{T})} \mathcal{L}(\theta_i; \mathcal{D}_i^q), \tag{3}$$

where $\beta$ is the learning rate for meta-updating. As a result of the meta-training phase, we get the well-generalized initialization $\theta_0^*$ for the base learner. In the meta-testing phase, the specific model $\theta_t$ for each target assay $\mathcal{T}_t$ with the support set $\mathcal{D}_t^s$ is achieved by a few gradient steps starting from the learned initialization $\theta_0^*$, i.e., $\theta_t = \theta_0^* - \alpha \nabla_\theta \mathcal{L}(\theta_0^*; \mathcal{D}_t^s)$. Finally, the performance is evaluated on the query set $\mathcal{D}_t^q$ of the target assay $\mathcal{T}_t$. Without loss of generality, we again take MAML as the backbone meta-learning framework of FRML and detail each component in the following.

### 3.2 Contrastive Assay Representation Learning

Learning the representation of assay $\mathcal{T}_i$ is a prerequisite to determining the functional regions that are specific to the assay. Following previous works [35, 37], we represent the assay with a representation vector $\mathbf{t}_i \in \mathbb{R}^d$ by aggregating all training samples of the support set $\mathcal{D}_i^s = \{(\mathbf{x}^s, \mathbf{y}^s)_i^j\}_{j=1}^{n_i^s}$, where an aggregator $\mathrm{AGG}$ is involved. The aggregator consists of a mapping function denoted as $\mathrm{MF}$ (e.g., recurrent network, convolutional network) that first encodes each individual sample into a dense representation vector, and a sample-level mean pooling layer to summarize all samples to generate the assay representation $\mathbf{t}_i$. Note that the pooling guarantees the assay representation to be invariant of the permutation of samples. Formally, we define the aggregation process as,

$$\mathbf{t}_i = \mathrm{AGG}(\mathcal{D}_i^s) = \frac{1}{n_i^s} \sum_{j=1}^{n_i^s} \mathrm{MF}(\mathcal{F}(\mathbf{x}_i^j) \oplus \mathbf{y}_i^j), \tag{4}$$

where $\mathcal{F}(\cdot)$ is an embedding function that transforms the input features into a low-dimensional vector. Both the embedded input features and the label are concatenated by the operator $\oplus$. We will provide more details on the definitions of $\mathcal{F}(\cdot)$ and $\mathrm{MF}(\cdot)$ later in Section 4.

The loss function to train the parameters $\mathcal{F}(\cdot)$ and $\mathrm{MF}(\cdot)$ could be Eqn. (3) only. Unfortunately, it is far from enough to learn a robust assay representation: first, the gradients back-propagated through the base learner and the region localization network tend to be too small for training to work effectively; second, the assay representation and the region localization are interleaving, so that the objective in Eqn. (3) takes them as a whole regardless of the accuracy for each of them. To overcome this limitation, we are motivated to impose another loss function on the assay representation network directly. The key intuition is that each set of samples in an assay provides a partial view of the assay, and the assay representation is expected to be consistent across views. This motivates the contrastive objective – different views of the same assay have similar task representations, while the representations of views from different assays should be different. Specifically, we create different views of assay $\mathcal{T}_i$ by randomly splitting $\mathcal{D}_i^s$ into $n_c$ sets of size $n_i^s/n_c$. By defining $c_u := ((u-1)n_i^s/n_c, \cdots, un_i^s/n_c)$, we obtain $n_c$ subsets of equal size, i.e., $\mathcal{D}_i^s = \cup_{u=1}^{n_c} \mathcal{O}_i^{c_u}$. We can now formulate the contrastive learning objective as follows:

$$\mathcal{L}_{cl} = \sum_{i=1}^{N} \sum_{1 \le u \le v \le n_c} \left[ \log \frac{\exp\left(\Phi\left(\mathrm{AGG}(\mathcal{O}_i^{c_v}), \mathrm{AGG}(\mathcal{O}_i^{c_u})\right)\right)}{\sum_{e=1}^{N} \exp\left(\Phi\left(\mathrm{AGG}(\mathcal{O}_i^{c_v}), \mathrm{AGG}(\mathcal{O}_e^{c_u})\right)\right)} \right], \tag{5}$$

where $\Phi$ is a similarity measure function. In our experiments, we adopt the dot product, i.e., $\Phi(a, b) = a^T b$. This contrastive loss function pushes the representations of different assays apart and meantime stabilizes the assay representation.

## 3.3 Localization Strategy

The assays are measured by different experimenters on different equipment, so that they are expected to have widely distributed assay representations. Given an assay with its representation, in this section, the localization strategy sets out to locate and assemble the functional regions that are specific to this assay. Before detailing the localization strategy, we first dissect the initialization $\theta_0$ of the base learner into $K$ functional regions. These functional regions are dissected in a layer-wise manner to maintain the hierarchical structure of the neural network. For each layer $l$, we denote its corresponding functional regions as $\theta_0^l = \{\theta_{0m^l}^l\}_{m^l=0}^{M^l}$, where $M^l$ represents the total number of functional regions in the $l$-th layer and $\sum_l M^l = K$.

Following the hierarchical representation in neural networks, we locate and assemble these functional regions in a hierarchical manner – each functional region at layer $l + 1$ receives signals from the functional regions at layer $l$. For each assay $\mathcal{T}_i$, denoting the representation of functional region $m^l$ in layer $l$ as $\mathbf{h}_i^{m^l}$, we define the representation of functional region $m^{l+1}$ in layer $l + 1$ to be:

$$\mathbf{h}_i^{m^{l+1}} = f^{m^{l+1}}(\sum_{m^l=1}^{M^l} p_i^{m^l \to m^{l+1}} \mathbf{h}_i^{m^l}), \quad \sum_{m^l=1}^{M^l} p_i^{m^l \to m^{l+1}} = 1, \tag{6}$$

where $f^{m^{l+1}}(\cdot)$ represents the mapping function for functional region $m^{l+1}$. $p_i^{m^l \to m^{l+1}}$ defined as the probability of functional region $m^l$ being assembled to $m^{l+1}$ is crucial; a value of $p_i^{m^l \to m^{l+1}} = 1$ suggests that functional region $m^l$ should be included for assay $\mathcal{T}_i$. Obviously, the probability $p_i^{m^l \to m^{l+1}}$ varies from assay to assay, so that we model it as a function of the representation $\mathbf{t}_i$, i.e.,

$$p_i^{m^l \to m^{l+1}} = \text{RG}(\mathbf{t}_i), \tag{7}$$

where $\text{RG}(\cdot)$ represents the region localization network we detail in the next subsection.

## 3.4 Region Localization Network

An ideal region localization network is expected to satisfy two criteria, including high representational capacity and consistency with the hierarchical structure behind functional regions. To meet the criteria, we propose a recurrent region localization network, where a recurrent neural network (GRU as exemplary) is used. The input to the recurrent neural network at step $l + 1$ is the combination of assay representation $\mathbf{t}_i$ and the assembly probabilistic set $\mathbf{p}_i^l$ of layer $l$, where $\mathbf{p}_i^l = \{p_i^{m^{l-1} \to m^l} | m^{l-1} \in [1, M^{l-1}], m^l \in [1, M^l]\}$. Consequently, the hidden representation at step $l + 1$ is,

$$\tilde{\mathbf{r}}_i^{l+1} = \text{GRU}(\mathbf{t}_i \oplus \mathbf{p}_i^l; \mathbf{r}_i^l)\mathbf{W}_f + \mathbf{b}_f, \tag{8}$$

where $\mathbf{W}_f \in \mathbb{R}^{d' \times M^l M^{l+1}}$ and $\mathbf{b}_f \in \mathbb{R}^{1 \times M^l M^{l+1}}$ are learnable parameters and $\tilde{\mathbf{r}}_i^{l+1} = \{\tilde{r}_i^{m^l \to m^{l+1}} | m^l \in [1, M^l], m^{l+1} \in [1, M^{l+1}]\} \in \mathbb{R}^{1 \times M^l M^{l+1}}$. The hidden representations at step $l + 1$, in return, determine the assembly probability at layer $l + 1$. Note that the assembly probability is expected to be as close to the bounds of its range $(0, 1)$ as possible, so that only the most pertinent functional regions are located. To this end, we apply the Gumbel-softmax estimator [10, 18] which models the categorical distribution to $\tilde{\mathbf{r}}_i$, i.e.,

$$p_i^{m^l \to m^{l+1}} = \frac{\exp((\tilde{r}_i^{m^l \to m^{l+1}} + q_i^{m^l \to m^{l+1}})/\tau)}{\sum_{s^l=1}^{M^l} \exp((\tilde{r}_i^{s^l \to m^{l+1}} + q_i^{s^l \to m^{l+1}})/\tau)}, \tag{9}$$

where $\tau$ is the temperature and $q_i^{m^l \to m^{l+1}}$ is sampled from the Gumbel distribution, i.e., $q_i^{m^l \to m^{l+1}} \sim \text{Gumbel}(0, 1)$.

Combining the meta-learning loss in Eqn. (3) and the contrastive loss in Eqn. (5), we arrive at the overall objective function of FRML defined as:

$$\min_{\Theta} \mathcal{L}_{all} = \min_{\Theta} \sum_{\mathcal{T}_i \in p(\mathcal{T})} \mathcal{L} + \lambda \mathcal{L}_{cl}, \tag{10}$$

where the hyperparameter $\lambda$ balances between two losses and $\Theta$ represents all learnable parameters. For better understanding of our framework, we show the meta-training process in Algorithm 1 and the meta-testing process in Appendix A.

---

**Algorithm 1** Meta-training Process of FRML

---

**Require:** $\{M^1, \ldots, M^L\}$: # of functional regions of each layer; $\alpha$, $\beta$: learning rates; $\lambda_1$, $\lambda_2$: item factors in loss

1: Randomly initialize $\Theta$
2: **while** not done **do**
3:     Sample a batch of assays from $p(\mathcal{T})$
4:     **for all** $\mathcal{T}_i$ **do**
5:         Sample $\mathcal{D}_i^s$, $\mathcal{D}_i^q$ from $\mathcal{T}_i$
6:         Get assay representation $\mathbf{t}_i$ in Eqn. (4) and the reconstruction loss $\mathcal{L}_{cl}$ via Eqn. (5)
7:         Use Eqn. (8) to compute $\{\tilde{\mathbf{r}}_i^1, \ldots, \tilde{\mathbf{r}}_i^L\}$
8:         Calculate $\{\mathbf{p}_i^1, \ldots, \mathbf{p}_i^L\}$ by Eqn. (9) and get the assembled trace across functional regions
9:         Use gradient descent to update parameters based on the learned trace: $\theta_i = \theta_0 - \alpha \nabla_\theta \mathcal{L}(\theta; \mathcal{D}_i^s)$
10:     **end for**
11:     Update $\Theta \leftarrow \Theta - \beta \frac{1}{N} \nabla_\Theta \sum_{\mathcal{T}_i \in p(\mathcal{T})} \mathcal{L}(\theta_i; \mathcal{D}_i^q) + \lambda \mathcal{L}_{cl}(\mathcal{D}_i^s)$
12: **end while**

---

## 4 Experiments

In this section, we empirically evaluate the effectiveness of FRML on two diverse drug discovery tasks: drug activity prediction and ADMET property prediction. We consider comparison of the proposed FRML with three categories of baselines. The first category simply using base learner without assay adaptation, including FC-Individual and FC-All. The second category is knowledge transfer with assay adaptation: Fine-tuning, MAML [6], ANIL [24], ANIL++ [3]. In the second category of methods, we also compared FRML with Prototypical Network (ProtoNet) [31] and Matching Network (MatchingNet) [34] for classification tasks, i.e., ADMET property prediction. The last category is heterogeneous meta-learning methods, including MMAML [35], HSML [37], ARML [38]. Detailed descriptions of all baselines are provided in Appendix B and the detailed hyperparameters for both applications are listed in Appendix D.

### 4.1 Drug Activity Prediction

#### 4.1.1 Dataset Description

For drug activity prediction, we use the dose-response activity assays from ChEMBL[1], where 4,276 assays are selected in this problem. Here, we randomly sample 100 assays as the meta-testing set, 76 assays as the meta-validation set, and the rest of assays for meta-training. The random splitting is repeated four times to construct four assay groups, named Assay Group I, II, III, IV, respectively. A few support and query drug compounds are available for each assay. In terms of the features for each drug compound, we use 1,024-dimensional Moragn fingerprint implemented in RDKit [12]. For each assay $\mathcal{T}_i$, we calculated the coefficient of determination ($R^2$) between the predicted value $\hat{\mathbf{Y}}_i^q$ and the ground truth value $\mathbf{Y}_i^q$. The median and mean $R^2$ values of all meta-testing assays are reported. We adopt another widely used metric for evaluating whether a virtual screening model is usable in practice, i.e., the number of assays with $R^2 > 0.3$. More detailed information and data statistics are summarized in Appendix C.1.

#### 4.1.2 Overall Performance

The performance of FRML and the baselines are reported in Table 1. In this experiment, FRML incorporates ANIL and ANIL++, while all other heterogeneous meta-learning algorithms (e.g., MMAML) incorporate ANIL. Note that ANIL++ is modified from ANIL to improve stability. From the results in Table 1, we obtain the key observations: (1) The performance of FC-Individual is inferior to that of other methods, indicates that involving the data from source assays benefits the performance; (2) Gradient-based meta-learning methods (MAML, ANIL, ANIL++, heterogeneous methods, and FRML) achieve significantly better performance than Fine-tuning, corrugating our motivation that Fine-tuning may confuse the most similar assays to the target with the others; (3) In most cases, heterogeneous methods (MMAML-ANIL, HSML-ANIL, ARML-ANIL, FRML-ANIL) achieve better performance than homogeneous meta-learning models, showing the effectiveness of integrating assay-specific knowledge transfer; (4) Our proposed FRML-ANIL++ achieves the best per-

Table 1: Performance of drug activity prediction (Measured by mean $R^2$, median $R^2$ and $\#R^2 > 0.3$).

| Model | Assay Group I | | | Assay Group II | | | Assay Group III | | | Assay Group IV | | |
|---|---|---|---|---|---|---|---|---|---|---|---|---|
| | Mean | Med. | $R^2$ >0.3 | Mean | Med. | $R^2$ >0.3 | Mean | Med. | $R^2$ >0.3 | Mean | Med. | $R^2$ >0.3 |
| FC-Individual | 0.141 | 0.064 | 16 | 0.114 | 0.060 | 10 | 0.112 | 0.046 | 10 | 0.118 | 0.047 | 10 |
| FC-All | 0.228 | 0.131 | 30 | 0.187 | 0.103 | 23 | 0.199 | 0.103 | 28 | 0.252 | 0.160 | 35 |
| Fine-tuning | 0.251 | 0.166 | 37 | 0.197 | 0.124 | 24 | 0.219 | 0.121 | 31 | 0.266 | 0.194 | 37 |
| MAML | 0.291 | 0.182 | 38 | 0.232 | 0.158 | 29 | 0.265 | 0.191 | 36 | 0.302 | 0.256 | 46 |
| ANIL | 0.299 | 0.184 | 41 | 0.226 | 0.143 | 30 | 0.268 | 0.199 | 37 | 0.304 | 0.282 | 48 |
| ANIL++ | 0.367 | 0.299 | 50 | 0.315 | 0.252 | 43 | 0.335 | 0.289 | 48 | 0.362 | 0.324 | 51 |
| MMAML-ANIL | 0.292 | 0.205 | 42 | 0.231 | 0.154 | 31 | 0.276 | 0.187 | 37 | 0.308 | 0.260 | 46 |
| HSML-ANIL | 0.295 | 0.192 | 41 | 0.234 | 0.145 | 34 | 0.277 | 0.196 | 35 | 0.306 | 0.254 | 47 |
| ARML-ANIL | 0.299 | 0.204 | 43 | 0.233 | 0.159 | 32 | 0.270 | 0.191 | 39 | 0.311 | 0.267 | 46 |
| **FRML-ANIL (ours)** | 0.310 | 0.226 | 44 | 0.237 | 0.162 | 35 | 0.285 | 0.207 | 40 | 0.322 | 0.287 | 49 |
| **FRML-ANIL++ (ours)** | **0.375** | **0.328** | **52** | **0.327** | **0.311** | **51** | **0.345** | **0.315** | **51** | **0.372** | **0.349** | **56** |

Table 2: Ablation study on drug activity prediction.

| Model | Assay Group I | | | Assay Group II | | | Assay Group III | | | Assay Group IV | | |
|---|---|---|---|---|---|---|---|---|---|---|---|---|
| | Mean | Med. | $R^2$ >0.3 | Mean | Med. | $R^2$ >0.3 | Mean | Med. | $R^2$ >0.3 | Mean | Med. | $R^2$ >0.3 |
| ANIL++ | 0.367 | 0.299 | 50 | 0.315 | 0.252 | 43 | 0.335 | 0.289 | 48 | 0.362 | 0.324 | 51 |
| Ablation I (w/o cl) | 0.371 | 0.315 | 51 | 0.318 | 0.263 | 45 | 0.338 | 0.305 | 49 | 0.368 | 0.338 | 54 |
| Ablation III (w/o localization) | 0.369 | 0.301 | 50 | 0.317 | 0.263 | 47 | 0.336 | 0.291 | 49 | 0.368 | 0.329 | 53 |
| Ablation II (RNN -> FC) | 0.372 | 0.303 | 52 | 0.326 | 0.299 | 50 | 0.341 | 0.306 | 50 | 0.367 | 0.333 | 53 |
| **FRML-ANIL++ (ours)** | **0.375** | **0.328** | **52** | **0.327** | **0.311** | **51** | **0.345** | **0.315** | **51** | **0.372** | **0.349** | **56** |

formance in all four assay groups. This possibly results from that differentiating neural sub-networks reduces the parameter space, which improves the generalization capability and further benefits the performance. Besides, integrating FRML with ANIL also achieves consistent improvements, showing its compatibility with different backbone meta-learning models.

### 4.1.3 Ablation Study

To further show the effectiveness of the proposed modules in FRML, we conduct comprehensive ablation studies by comparing FRML with three ablation models described as follows. First, we consider an ablation model (**Ablation I (w/o cl)**) with the contrastive loss removed. Second, we design **Ablation II (w/o localization)** to show that the improvements of FRML is caused by knowledge localization rather than increasing the capacity of baseline. Third, we change the recurrent structure to a plain localization network and propose **Ablation III (RNN->FC)**, where fully connected layers with softmax are utilized to learn the assembly probability set $\{\mathbf{p}_i^1, \ldots, \mathbf{p}_i^L\}$.

We evaluate the ablation models on all four assay groups and report the performance in Table 2. Note that FRML is also included in comparison. From the results in the table, we have the following three findings: (1) removing the contrastive loss hurts the performance, which indicates the effectiveness of the contrastive loss in learning well-differentiated assay representations; (2) the superiority of FRML over abalation II demonstrates that the improvements stem from efficient knowledge structuring rather than larger model capacity; (3) compared to the plain localization network, the performance gain of recurrent region localization network demonstrates its superiority by predicting the assembly probability in a hierarchical way.

### 4.1.4 Analysis

**Effect of the Number of Functional Regions.** We analyze the effect of the number of functional regions and illustrate the results in Figure 3(a). In this figure, we observe that (1) if the number of functional regions is too small (e.g. 1), it may be insufficient to capture the structures across assays. (2) when we continually increase the number of functional regions, the results keep stable or even slightly decrease, which are consistent with our find-

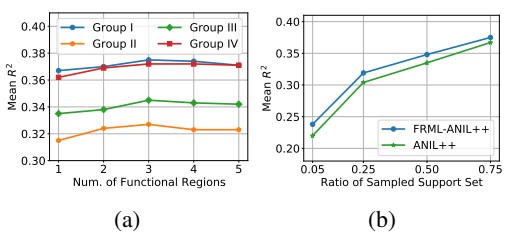

(a)        (b)

Figure 3: (a): Num. of functional regions w.r.t. the mean $R^2$ on Assay Group I, II, III, IV. (b): Performance w.r.t. support set ratio on Assay Group I.

ings that the gains of FRML arise out of the effective knowledge structuring instead of the increase of the model capacity.

Besides, we also conduct an experiment using a different number of functional regions in each layer. Here, we use two functional blocks in layer one and four functional blocks in layer two. The results are reported in Table 3. We observe that the performance is slightly worse than the strategy of using three layers for each layer. One potential reason is that the Morgan fingerprint features have covered enough low-level features, and it might be more useful to add more functional blocks in the first layer.

Table 3: Performance of FRML with different knowledge blocks in each layer on drug activity prediction. (2, 4) represents that we use 2 blocks in layer 1 and 4 blocks in layer 2. G-I represents group I. Mean $R^2$ is reported.

| # of Blocks | G-I | G-II | G-III | G-IV |
|---|---|---|---|---|
| (2, 4) | 0.369 | 0.324 | 0.344 | 0.367 |
| **(3, 3)** | **0.375** | **0.327** | **0.345** | **0.372** |

**Effect of the Ratio of the Support set.** In order to show the superiority of FRML under different ratios of the support set, we analyze the performance w.r.t. the support set ratio and show the results in Figure 3(b). When we down-sample the support set to contain $5\%$, $25\%$, and $50\%$ of all compounds in an assay, FRML consistently achieves better performance than the most competitive baseline ANIL++. This marks the capability of FRML in handling the data scarcity problem in healthcare.

**Analysis of Localization Strategy.** We further analyze the localization strategy, where the assembled traces of six randomly selected meta-testing assays from Group II are illustrated in Figure 4(a)-(f) and their corresponding biological properties are reported in the right table of Figure 4. Here, we observe that the six assays are mainly located in three different traces. Besides, assays 1640791, 701282, 1639959, 302952 activate the same trace 1→21. The trace groups are consistent with their biological properties reported in the table. First, assays 1640791 and 701282 are both cell-based functional assays targeting GPCRs by evaluating the antagonistic activity of compounds to their downstream cAMP pathway. 1639959 and 302952 are both cell-based functional assay of membrane transporters. All four assays are targeting membrane proteins (receptors or transporters). Thus, they share the first layer but select different traces in the second layer. Second, different from the above four assays, assays 147797 and 1520 choose a completely different path since they are single protein assays that directly evaluate the effect of compounds to their protein targets. The consistency of localization results and biological properties further verify the effectiveness of FRML for distinguishing different domains via localization strategy.

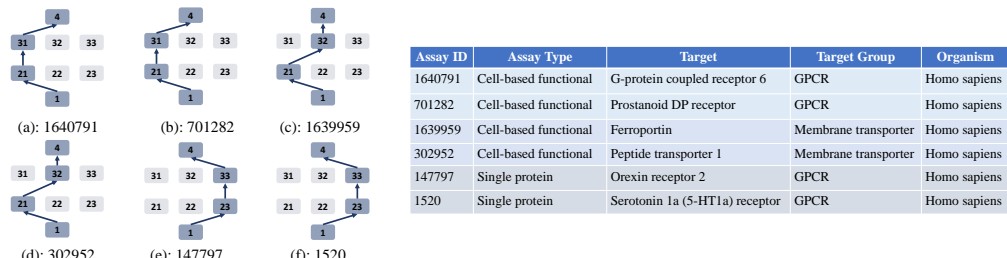

Figure 4: Left Figure (a)-(f) show the located traces from six meta-testing assays of Group II, where their corresponding biological information are reported in the right table. Darker blocks and blue links represent located functional regions and assembled links, respectively.

## 4.2 ADMET Property Prediction

### 4.2.1 Dataset Description & Evaluation Metric

Besides the drug activity prediction, we further evaluate FRML on ADMET property prediction. The AMDET Prediction problem is constructed by combining 4 benchmark datasets from the MoleculeNet [36] with biophysiology and physiology targets. The 4 datasets are MUV [29], SIDER [11], Tox21 and ToxCast [27]. Each property prediction is a binary classification task. All the properties from MUV, SIDER, Tox21, and 22 properties form ToxCast are involved in the experiment, resulting in 68 tasks. We randomly sample 42 tasks for meta-training and use the remaining 26 tasks for meta-testing. Considering the data balance, for each tasks, we randomly sample only partial instances from the majority category to match the size of minority data, together with all the minority data, to

Table 4: Performance of ADEMT property prediction (averaged accuracy with 95% confidence interval are reported).

| Model | SIDER | Tox21 | MUV | ToxCast |
|---|---|---|---|---|
| FC-Individual | $52.12 \pm 0.81\%$ | $51.25 \pm 0.37\%$ | $52.91 \pm 0.67\%$ | $62.75 \pm 1.27\%$ |
| FC-All | $67.13 \pm 0.89\%$ | $68.63 \pm 0.84\%$ | $55.04 \pm 1.06\%$ | $70.82 \pm 1.61\%$ |
| Fine-tuning | $67.60 \pm 0.89\%$ | $68.84 \pm 0.84\%$ | $55.41 \pm 1.05\%$ | $71.04 \pm 1.59\%$ |
| MAML | $67.69 \pm 0.81\%$ | $69.12 \pm 0.84\%$ | $56.66 \pm 1.09\%$ | $72.53 \pm 1.64\%$ |
| ProtoNet | $68.03 \pm 1.16\%$ | $69.29 \pm 1.30\%$ | $55.19 \pm 1.18\%$ | $72.10 \pm 1.52\%$ |
| MatchingNet | $66.69 \pm 1.04\%$ | $68.72 \pm 1.04\%$ | $55.15 \pm 1.07\%$ | $71.39 \pm 1.33\%$ |
| ANIL | $67.92 \pm 0.89\%$ | $69.81 \pm 0.85\%$ | $55.13 \pm 1.22\%$ | $72.09 \pm 1.78\%$ |
| ANIL++ | $68.04 \pm 0.86\%$ | $68.94 \pm 0.92\%$ | $56.95 \pm 1.13\%$ | $72.66 \pm 1.67\%$ |
| MMAML-ANIL | $68.57 \pm 0.82\%$ | $69.86 \pm 0.90\%$ | $58.06 \pm 1.21\%$ | $72.10 \pm 1.55\%$ |
| HSML-ANIL | $69.15 \pm 0.87\%$ | $69.98 \pm 0.88\%$ | $57.94 \pm 1.18\%$ | $71.73 \pm 1.46\%$ |
| ARML-ANIL | $68.94 \pm 0.84\%$ | $70.07 \pm 0.91\%$ | $58.99 \pm 1.16\%$ | $72.08 \pm 1.56\%$ |
| **FRML-ANIL (ours)** | $69.89 \pm 0.87\%$ | $70.85 \pm 0.85\%$ | $59.94 \pm 1.00\%$ | $73.56 \pm 1.58\%$ |
| **FRML-ANIL++ (ours)** | $\mathbf{70.01 \pm 0.86}\%$ | $\mathbf{71.07 \pm 0.91}\%$ | $\mathbf{60.66 \pm 1.09}\%$ | $\mathbf{74.02 \pm 1.57}\%$ |

form the task dataset. In this experiment, following the conventional few-shot learning protocol [7], we apply 2-way classification with 5-shot support samples for each task. The details of the dataset descriptions are available in Appendix C.2. As for the model performance, it is measured by averaged classification accuracy.

### 4.2.2 Results

We report the performance of FRML and the baselines in Table 4. Similar findings to that of drug activity prediction experiments are observed. Therefore, we again confirm the effectiveness and importance of integrating task-specific knowledge transfer in the proposed FRML. A specific finding is that: all heterogeneous meta-learning models and FRML obtain higher gain of performance on MUV than on the three datasets. In particular, FRML achieves significant performance improvement on MUV dataset. This may be caused by the category difference of MUV from the other three datasets which we will detail in the next subsection. Besides, we conduct similar ablation studies to those for drug activity prediction and report the results in Table 5. Similar results are observed, again demostrating the effectiveness of FRML in differentiating different properties.

Table 5: Ablation study of ADEMT property prediction (averaged accuracy with 95% confidence interval are reported).

| Model | SIDER | Tox21 | MUV | ToxCast |
|---|---|---|---|---|
| ANIL++ | $68.04 \pm 0.86\%$ | $68.94 \pm 0.92\%$ | $56.95 \pm 1.13\%$ | $72.66 \pm 1.67\%$ |
| Ablation I (w/o cl) | $68.79 \pm 0.84\%$ | $69.65 \pm 0.86\%$ | $58.70 \pm 1.18\%$ | $72.86 \pm 1.68\%$ |
| Ablation II (w/o localization) | $68.30 \pm 0.83\%$ | $69.42 \pm 0.92\%$ | $57.33 \pm 1.04\%$ | $72.70 \pm 1.55\%$ |
| Ablation III (RNN->FC) | $69.15 \pm 0.80\%$ | $70.08 \pm 0.87\%$ | $59.54 \pm 1.00\%$ | $73.41 \pm 1.68\%$ |
| **FRML-ANIL++ (ours)** | $\mathbf{70.01 \pm 0.86}\%$ | $\mathbf{71.07 \pm 0.91}\%$ | $\mathbf{60.66 \pm 1.09}\%$ | $\mathbf{74.02 \pm 1.57}\%$ |

### 4.2.3 Analysis of Localization Strategy

In this part, we analyze the localization strategy for ADMET prediction. In Figure 5, we show the assembled traces of four meta-testing tasks sampled from different sub-datasets. In these figures, tasks from different sub-domains are located in different trace groups (i.e., the three datasets SIDER, Tox21, Tox-cast select 1→21→32→4 while MUV selects

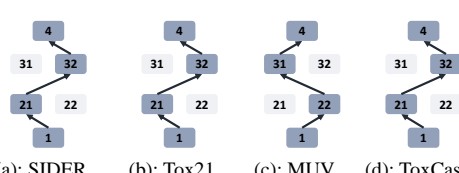

(a): SIDER    (b): Tox21    (c): MUV    (d): ToxCast

Figure 5: (a)-(d) show the assembled traces (blue links) among located regions (darker blocks) from four meta-testing tasks sampled from SIDER, Tox21, MUV, ToxCast, respectively.

1→22→31→4, respectively). Compared to SIDER, Tox21, Toxcast, we notice that MUV selects a different trace, which matches the natural difference between MUV and the other three datasets. The category of the MUV dataset is a biophysics while that of the other three are physiology. Besides, MUV is designed for validation of virtual screening techniques, while the other three are designed for measuring different targets.

## 5 Related Work

The goal for meta-learning is to learn a set of meta-knowledge that facilitates the learning process of new tasks. There are two mainstream categories of meta-learning approaches. The first category of algorithms, called gradient-based meta-learning algorithms, regards the meta-knowledge as initializations for the base learner [7–9, 13, 14, 19, 25, 30]. As for the second category, i.e., metric-based meta-learning algorithms, the aim is to learn a transferable metric space for the meta-learner as well as a lazy learner [15, 31, 32, 34, 39]. However, metric-based algorithms only handle classification problems. In light of this, we consider gradient-based algorithms which are flexible and general enough to be independent of problem types. The majority of gradient-based meta-learning algorithms focus on maintaining a shared set of meta-knowledge (i.e., the initializations for the weights) learned from meta-training tasks. To enhance the ability of generalization to more complicated heterogeneous tasks (e.g., tasks sampled from various distributions), recent studies customize the shared model weight initializations to different tasks modulating the globally-shared weight initializations to be task-specific [20, 35, 37, 38]. However, our proposed FRML goes further than customization of weight initializations – it also differentiates neural sub-networks and enhances the generalization capability for significantly different tasks.

Up to now, only a few studies have explored the application of meta-learning to address the problem of limited labeled data in healthcare. The two representative metric-based meta-learning algorithms, i.e., MatchingNet [34] and ProtoNet [31], have been used for protein binding prediction [2] and dematological disease diagnosis [22]. As we mentioned above, metric-based meta-learning algorithms do not work for the regression of activity values we focus on in this work. On the other hand, Zhang et al. [40], Qiu et al. [23], and Luo et al. [17] applied the widely used model agnostic meta-learning (MAML) algorithm [7] to the problems of clinical risk prediction, genomic survival analysis, and protein binding, respectively. Yet, the proposed FRML accommodates a wide range of assays effectively by tailoring sub-networks for each assay.

## 6 Conclusion

In this paper, we aim to tackle the challenge of data insufficiency in drug discovery by transferring the knowledge from historical assays. Specifically, we propose a novel meta-learning framework, FRML, to effectively learn the transferable knowledge and meantime adapt to various assays. FRML dissects the base learner into hierarchically organized functional regions. The representation of a target assay is forwarded to the recurrent region localization network to locate and assemble the assay-specific functional regions. The experiments on virtual screening and ADMET prediction demonstrate the effectiveness of FRML, and the analyses onThe the localization strategy further verify its sound interpretability in capturing the similarity between assays.

The limitation of this work is that we have not investigated the robustness of the proposed FRML. If the proposed framework is easy to be attacked, it may cause negative social impacts. For example, if the framework suggests misleading results, it will delay even harm drug discovery progress. We will investigate the problem in the future.

### Acknowledgement

H.Y. and Z.L. are supported in part by NSF awards IIS-#1652525 and IIS-#1618448. The views and conclusions contained in this paper are those of the authors and should not be interpreted as representing any funding agencies. This work was also supported in part by the start-up grant from City University of Hong Kong (9610512). Besides, Y.W. would like to acknowledge the support from the Tencent AI Lab Rhino-Bird Gift Fund.

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
