# Appendix of Functionally Regionalized Knowledge Transfer for Low-resource Drug Discovery

**Huaxiu Yao**[1]*, **Ying Wei**[2], **Long-Kai Huang**[3], **Ding Xue**[3]
**Junzhou Huang**[4], **Zhenhui Li**[5]
[1]Stanford University, [2]City University of Hong Kong, [3] Tencent AI Lab
[4]University of Texas at Arlington, [5]Pennsylvania State University
[1]huaxiu@cs.stanford.edu, [2]yingwei@cityu.edu.hk, [3]hlongkai@gmail.com, dingxue@tencent.com
[4]jzhuang@uta.edu, [5]jessieli@ist.psu.edu

## A   Meta-testing Framework

We show the framework of meta-testing in Alg. 1.

## B   Detailed Baseline Descriptions

For FC-individual, we train each testing assay separately with a two-layer fully-connected base learner. For FC-All, a two-layer fully connected model is trained on samples from both support set and query set of source assays and from the support set of the target assay. Both FC-individual and FC-All directly apply the learned model on the query set of the target assays to predict the results. For Fine-tuning, we train a global-shared model on data from all source assays. Then, for the target assay, we fine-tune the learned model on the support set of the target assay, and then evaluate the performance on the corresponding query set. As for all meta-learning algorithms (MAML, ANIL, ANIL++, and heterogeneous meta-learning methods), we apply the same meta-training and meta-testing procedures described in Section 2. Note thatall the base learners in the baseline methods share the same structure.

## C   Data Statistics

### C.1   Drug Activity Prediction Data

For drug activity prediction, here we summarized the number of assays belonging to each target family: GPCR (685), Ion channel (215), Kinase (665), NHR (123), Binding (2523), Phenotypic (2299), Functional (1689), Proteinase (289), ADME (55). Then, we list the meta-validation assays and meta-testing assays for each group, and use the rest assays for meta-training.

- Assay Group I:
  - Meta-testing: 752640, 972801, 737284, 954885, 1528837, 1587725, 1527823, 1640977, 157713, 1285138, 1437208, 1349151, 1592870, 93228, 465460, 954934, 84556, 1567308, 1577550, 1285709, 654928, 620647, 864364, 575603, 1280627, 688257, 1443970, 1527947, 737424, 201877, 1457820, 603293, 809120, 883875, 1641128, 1534634, 1641655, 955073, 954571, 736971, 577227, 45264, 455393, 728290, 688357, 1301747, 105205, 865015, 665348, 820998, 759559, 1301769, 609034, 80649, 1641240, 965916, 34078, 1470241, 1348900, 333106, 1527607, 954703, 1641298, 1641300, 727385, 304989, 981861, 212325, 756584, 331630, 473976, 63356, 51590, 1640328, 954762, 1642379, 1527698, 1527704,

---
*Part of the work was done when H.Y. was a student at Penn State University, correspondence to: Y.W.

**Algorithm 1** Meta-Testing of FRML

---

**Require:** $\mathcal{D}_t^s$: support data of the target assay $\mathcal{T}_t$; $\mathcal{D}_t^q$: query data of the target assay $\mathcal{T}_t$; $\alpha$: inner learning rate; $\Theta^*$: optimized model parameters during the meta-training process
1: Compute the task representation $\mathbf{t}_t$ via Eqn. (4)
2: Compute $\{\tilde{\mathbf{r}}_t^1, \ldots, \tilde{\mathbf{r}}_t^L\}$ via Eqn. (9)
3: Calculate $\{\mathbf{p}_t^1, \ldots, \mathbf{p}_t^L\}$ by Eqn. (10) and generate the assembled trace across functional regions
4: Update parameters $\theta_t = \theta_0^* - \alpha \nabla_\theta \mathcal{L}(\theta; \mathcal{D}_t^s)$
5: Evaluate the performance using $\theta_t$ and $\mathcal{D}_t^q$

---

543133, 954781, 1301405, 619939, 605612, 585134, 1433006, 934321, 1642435, 1637320, 936907, 54735, 70610, 1508820, 1292758, 104407, 992729, 199642, 160234, 1528304, 629753, 931327

  – Meta-validation: 972800, 688641, 610565, 1536390, 211079, 1625735, 1641357, 688654, 1641103, 457234, 450707, 195220, 1366808, 49308, 924, 49312, 828065, 737313, 1528100, 596645, 1641767, 1535401, 688427, 969260, 453677, 978479, 1641008, 574385, 911154, 446257, 878513, 1640955, 902584, 1276473, 752567, 306492, 736957, 1640384, 1454018, 2755, 579907, 1527622, 761927, 89542, 809158, 978889, 556876, 478840, 688464, 1330005, 144341, 1528791, 1301597, 1641310, 209245, 608993, 1528801, 89064, 1527913, 4202, 688616, 1513, 510189, 1641197, 1527791, 688495, 89839, 1641201, 1528688, 752371, 688379, 938230, 596087, 835704, 566779, 688767

- Assay Group II:
  – Meta-testing: 835072, 539657, 1641997, 1639955, 1638422, 1639959, 1622038, 637980, 28188, 91168, 954915, 425511, 688685, 155185, 39493, 155208, 1641035, 1288277, 755797, 954462, 812132, 87656, 1536113, 48248, 744057, 210045, 1642144, 50337, 325795, 1527974, 1642150, 814256, 1641143, 438974, 217297, 1641170, 688340, 1641688, 688357, 649964, 930033, 447747, 566532, 1641737, 49425, 562451, 817939, 688403, 817944, 52506, 452895, 984872, 311595, 899888, 646978, 1642307, 664904, 1641802, 1466703, 1466704, 809297, 147797, 1640791, 305497, 209245, 603488, 701282, 752485, 302952, 122731, 563052, 1561972, 1528692, 1642361, 1528698, 737150, 1301374, 51590, 364426, 1642378, 899993, 752538, 1640355, 1446827, 62394, 842684, 1640893, 44489, 688589, 208335, 1642449, 858065, 1640919, 1528791, 1528294, 1520, 1640952, 92156, 63997, 69119
  – Meta-validation: 7296, 1276546, 87173, 688645, 1350406, 955016, 697223, 1163, 201739, 809231, 1528850, 1528212, 752533, 971798, 954388, 1626011, 1528480, 501795, 1527972, 470053, 1640867, 809128, 737064, 1642538, 954282, 978478, 786095, 29233, 1642418, 737075, 1536179, 1641399, 1527735, 609465, 1640506, 1641659, 307259, 1537597, 769089, 140229, 789189, 860488, 766795, 48587, 1528909, 1451727, 219472, 737105, 955090, 311637, 1528022, 1632983, 727385, 1456602, 1641179, 688347, 67039, 434528, 1564001, 727521, 688483, 595939, 1436004, 736997, 1528160, 1640426, 4202, 102381, 45422, 1641073, 47858, 37363, 1641720, 688889, 1301756, 556797

- Assay Group III:
  – Meta-testing: 688643, 688645, 159749, 1527820, 539663, 303638, 1638422, 954399, 330271, 70695, 1295917, 1642542, 1527862, 1528890, 200254, 540741, 1365575, 761928, 688201, 37966, 1537108, 336476, 511069, 1301599, 1528416, 206959, 478840, 1503357, 1642117, 1536654, 688274, 1527963, 688288, 688293, 688816, 745138, 438974, 88771, 459971, 714443, 1289425, 1451729, 1284820, 954602, 954604, 208118, 198910, 809216, 610565, 1528071, 453897, 770827, 216843, 828171, 306447, 562451, 1642271, 1528097, 28965, 367910, 1642296, 1528125, 1528145, 1555281, 49489, 493905, 876885, 1290079, 468834, 1528677, 756582, 1338728, 1528170, 845165, 1528696, 617338, 1301888, 102785, 1527682, 940424, 1528724, 809380, 1446827, 864186, 1291714, 642499, 688586, 1513931, 32721, 954834, 1536468, 688598, 1556442, 901084, 954845, 1527780, 1640932, 477677, 829947, 1528829
  – Meta-validation: 856700, 688641, 448646, 1613063, 1301767, 624014, 559247, 1527953, 1640339, 49558, 737046, 809242, 1642522, 1641371, 1527965, 592925, 954655, 688416, 305569, 538786, 1535011, 208672, 1592863, 688550, 1527974, 1527976, 1642272, 305065, 809259, 1640189, 96941, 688685, 954799, 978480, 934321, 1637168, 29233, 45236, 306221, 1535033, 1640506, 1290683, 158524, 936637, 647615, 422463, 1459648, 1640904, 954953,

1361352, 654923, 1641164, 954959, 1301583, 688210, 1508820, 45272, 688346, 737371, 162397, 775393, 535396, 1301477, 4197, 651627, 75756, 3819, 737391, 1641201, 1528692, 1294964, 456311, 737273, 1301756, 1640573, 42878

- Assay Group IV:
  - Meta-testing: 1556484, 688644, 737290, 1301520, 1642001, 1528850, 688661, 971799, 1642520, 46624, 1537067, 1641005, 1641010, 688185, 954938, 443966, 599616, 439367, 1537607, 954956, 688719, 615506, 654934, 1589851, 104542, 457824, 1641573, 1527916, 737391, 1301619, 211078, 52874, 955024, 32404, 158358, 566940, 50848, 1527970, 1349288, 1586856, 860330, 688818, 1536181, 48316, 491718, 1296583, 954567, 1566412, 66255, 66267, 809183, 425699, 467683, 464617, 752377, 508163, 872708, 1640197, 1301765, 809221, 809239, 1528102, 809255, 1536298, 1640747, 688431, 1636657, 213817, 1466703, 1301330, 1545042, 737622, 1452895, 950625, 856937, 493931, 954743, 809346, 558984, 1642378, 591251, 1640852, 305050, 934299, 1640862, 1640351, 1642418, 558515, 1511354, 1330619, 1528770, 1291715, 901575, 1640904, 1439182, 1537998, 737235, 1301469, 37371, 797692
  - Meta-validation: 688391, 1641992, 1641737, 306314, 311816, 813068, 68748, 1536775, 823822, 1639955, 696215, 1469079, 971801, 41884, 637980, 1298461, 76063, 688416, 1637151, 619938, 1642274, 885155, 572966, 510887, 1641000, 1528488, 954411, 1642415, 138287, 1527985, 56498, 27571, 458930, 311855, 809146, 307259, 950588, 736957, 1527742, 809152, 592450, 809156, 1528648, 954957, 209231, 490576, 1528273, 1641170, 45265, 1528917, 1528149, 1292759, 809175, 53367, 737370, 822749, 154333, 67039, 737273, 1640162, 737379, 763492, 809193, 954987, 104172, 510189, 1528695, 208754, 688243, 45044, 954482, 1640307, 1528183, 1642489, 1527674, 688254

## C.2 ADMET Data

All datasets involved in ADMET Prediction problem are public dataset [2]. The details of each dataset are listed below and data statistics are provided in Table 1:

- MUV [3] is a subset of PubChem BioAssay [4]. It is constructed by refined nearest neighbor analysis for validation of virtual screening.
- SIDER[1] contains records of marketed drugs along with its adverse drug reactions, also known as the Side Effect Resource.
- Tox21 is a dataset for 2014 Tox21 Data Challenge to measure the toxicity of compounds on 12 biological targets.
- ToxCast [2] contains 60 toxicity labels over thousands of compounds by running high-throughput screening tests on thousands of chemicals.

Table 1: Datasets Summary of ADMET Property Prediction.

| Category | Dataset | # of Sampled Tasks | # Tasks in Meta-Train | # Tasks in Meta-Test | # Samples |
|---|---|---|---|---|---|
| Biophysics | MUV | 17 | 11 | 6 | 48 to 60 |
| Physiology | SIDER | 27 | 16 | 11 | 44 to 1368 |
| | Tox21 | 12 | 8 | 4 | 372 to 1884 |
| | ToxCast | 22 | 15 | 7 | 50 to 1140 |

# D  Hyperparameters

In both drug activity prediction and ADMET property prediction, the preliminary mapping function $\mathcal{F}(\cdot)$ is similar to the base learner, which consists of two fully connect layer blocks with 500 neurons each layer. In drug activity prediction, we set the mapping function $\mathrm{MF}(\cdot)$ as one fully connected layers with 64 neuron. The number of functional regions is set as 3. The inner loop learning rate $\alpha$

---
[2]http://moleculenet.ai/datasets-1

and the outer loop learning rate $\beta$ are set as 0.005 and 0.001, respectively. In ADMET prediction, the mapping function $\text{MF}(\cdot)$ is set as one bi-directional LSTM with 32 neurons in each direction. We set the number of functional regions, the inner loop learning rate $\alpha$ and the outer loop learning rate $\beta$ as 2, 0.1, 0.01, respectively. The number of task adaptation optimization steps for both prediction problems is set as 5.