# OpenReview forum: "Functionally Regionalized Knowledge Transfer for Low-resource Drug Discovery"
_NeurIPS.cc/2021/Conference — NeurIPS 2021 Poster_

### Official Review · Reviewer_BVPN · 2021-07-14

**Rating:** 7
**Confidence:** 4

**Summary:**

This paper seeks to remedy the low-resource drug discovery problem by transferring the knowledge from previous assays, namely in-vivo experiments, by different laboratories and against various target proteins. The authors propose a functional rationalized meta-learning algorithm FRML for such knowledge transfer. The compositionality of the model improves the capacity of generalization to various and even out-of-distribution tasks. Empirical results on both virtual screening and ADMET prediction validate the superiority of FRML over state-of-the-art baselines powered with interpretability in assay relationships.

**Ethics Review Area:**

["I don’t know"]

**Limitations And Societal Impact:**

As the authors presented in the paper, the limitation of this work is that they have not investigated the robustness of the proposed FRML. If the proposed framework is easy to be attacked, it may cause negative social impacts.

**Main Review:**

This paper aims to tackle the challenge of data insufficiency in drug discovery by transferring the knowledge from historical assays, with a novel meta-learning framework, FRML, to effectively learn the transferable knowledge and meantime adapt to various assays.

Pros:

- Overall, the idea of this paper is very valuable for low-resource drug discovery, the originality is good and the authors also present the difference of this work with previous contributions.
- The quality of this paper is technically sound. Experimental results on virtual screening and ADMET prediction also show the effectiveness of FRML, and the analyses on the localization strategy further verify its sound interpretability in capturing the similarity between assays.
- The proposed FRML framework is reasonable and interesting for accommodating a wide range of assays effectively by tailoring sub-networks for each assay. This paper provides comprehensive experiments, including Drug Activity Prediction and ADMET Property Prediction, to show the effectiveness of the proposed model. The authors also carried out an ablation study to further show the effectiveness of the proposed modules in FRML.
- The paper is well-written and the design decisions are clearly explained. The comparison of benchmark methods is also interesting to read. In general, I think this is a worthy publication.


Cons:

- The choice of the baseline method. Why did the author choose these three categories of baselines? There are some other meta-learning methods, such as Non-parametric methods or Bayesian meta-learning. Why doesn't the author consider comparing these methods?
- The paper stated the proposed method improves the capacity of generalization to various and even out-of-distribution tasks. What is the out-of-distribution task? Is this a statistically significant distribution? The author needs to provide a more detailed explanation.
- Some minor typos:
  - line 373: "analyses onThe the localization strategy" -> analyses on the localization strategy
  - Appendix B line 12: "Note thatall the base learners" -> Note that all the base learners


**Time Spent Reviewing:**

1.0

---

> ### Author Response · Authors · 2021-08-10
> **Response to Reviewer BVPN**
>
> Thank you for your review and your valuable comments and suggestions. We address your comments point by point as follows. Could you please kindly let us know if you have any additional reservations, or whether our response adequately addresses your concerns.
>
> **Q1**: Choice of baseline methods
>
> **A1**: As discussed in Line 223-227, the reasons why we choose the three categories of baselines are to show
> - The performance of non-meta-learning methods as the **first category** of methods.
> - The performance of basic meta-learning baselines as the **second category** of methods. Specifically, as discussed in Line 348-351 (main paper), we use gradient-based meta-learning methods rather than non-parametric methods since the latter can only be used in classification problems.
> - The performance of heterogeneous meta-learning methods as the **third category** of methods. They are more powerful than basic meta-learning baselines and are the most relevant to FRML (see detailed discussion in Line 62-70 and Line 351-356 in the main paper).
>
> We follow the reviewer’s constructive suggestion to also investigate the performance of non-parametric and Bayesian meta-learning methods. Here, we select ProtoNet [1] and Matching Network [2] as two representative methods in non-parametric meta-learning, and BMAML [3] as the representative method for Bayesian meta-learning.
> - For drug activity prediction, which is a regression problem, the two non-parametric meta-learning methods are not applicable, and we compare with BMAML only. We show the results in Table Re-1, where FRML-ANIL++ capable of differentiating function regions for different tasks outperforms BMAML significantly.
> - For ADMET classification, which is a classification problem, we compare both the two non-parametric meta-learning methods and the Bayesian meta-learning method, i.e., ProtoNet, Matching Network, and BMAML. We show the results in Table Re-2, where again FRML-ANIL++ stands out.
>
>
> **Table Re-1**: Comparison between FRML with BMAML on drug activity prediction.
>
> |             | Group I |       |           | Group II |       |           | Group III |       |           | Group IV |       |           |
> |-------------|:-------:|:-----:|:---------:|:--------:|:-----:|:---------:|:---------:|:-----:|:---------:|:--------:|:-----:|:---------:|
> |             |   Mean  |  Med. | $R^2>0.3$ |   Mean   |  Med. | $R^2>0.3$ |    Mean   |  Med. | $R^2>0.3$ |   Mean   |  Med. | $R^2>0.3$ |
> | BMAML       |  0.302  | 0.194 |     37    |   0.228  | 0.128 |     31    |   0.273   | 0.183 |     38    |   0.305  | 0.230 |     44    |
> | **FRML-ANIL (ours)** |  0.310  | 0.226 |     44    |   0.237  | 0.162 |     35   |   0.285   | 0.207 |     40   |   0.322  | 0.287 |     49   |
> | **FRML-ANIL++ (ours)** | **0.375** | **0.328** | **52** | **0.327** | **0.311** | **51** | **0.345** | **0.315** | **51** | **0.372** | **0.349** | **56** |
>
> **Table Re-2**: Performance (accuracy $\pm$ 95% confidence interval) of non-parametric methods (ProtoNet, MatchingNet) and Bayesian meta-learning method (BMAML) on ADMET property classification.
>
> |             |       SIDER       |       Tox21       |        MUV        |      ToxCast      |
> |-------------|:-----------------:|:-----------------:|:-----------------:|:-----------------:|
> | ProtoNet    | 68.03 $\pm$ 1.16% | 69.29 $\pm$ 1.30% | 55.19 $\pm$ 1.18% | 72.10 $\pm$ 1.52% |
> | MatchingNet | 66.69 $\pm$ 1.04% | 68.72 $\pm$ 1.04% | 55.15 $\pm$ 1.07% | 71.39 $\pm$ 1.33% |
> | BMAML       | 67.86 $\pm$ 1.20% | 68.45 $\pm$ 0.85% | 57.78 $\pm$ 1.22% | 72.37 $\pm$ 1.65% |
> | **FRML-ANIL (ours)** | 69.89 $\pm$ 0.87% | 70.85 $\pm$ 0.85% | 59.94 $\pm$ 1.00% | 73.56 $\pm$ 1.58% |
> | **FRML-ANIL++ (ours)**  | **70.01 $\pm$ 0.86%** | **71.07 $\pm$ 0.91%** | **60.66 $\pm$ 1.09%** | **74.02 $\pm$ 1.57%** |
>
> ***
>
> **Q2**: Out of distribution tasks
>
> **A2**: An out-of-distribution task refers to a meta-testing task that may come from a distribution that is different from the task distribution the model has been trained on. One intuitive example is that the meta-training tasks are all about fine-grained bird classification, while an out-of-distribution meta-testing task is about flower classification. In our experiment of drug discovery, we conduct an additional experiment on ADMET property classification, where we train the model on SIDER, MUV, Tox21, and evaluate the performance on ToxCast. The results are reported in Table Re-3, and we observed that FRML indeed improves the generalization capacity on the out-of-distribution tasks.
>
> **Table Re-3**: Results (accuracy $\pm$ 95% confidence interval) of out-of-distribution generalization on ADMET property prediction.
>
> | Domains | Meta-training: SIDER, Tox21, MUV; Meta-testing: ToxCast |
> |----------------------|:-------------------:|
> | FC-Individual        |  62.75 $\pm$ 1.27%  |
> | FC-All               |  64.92 $\pm$ 1.29%  |
> | ANIL                 |  67.16 $\pm$ 1.33%  |
> | ANIL++               |  67.89 $\pm$ 1.52%  |
> | ARML                 |  67.93 $\pm$ 1.44%  |
> | **FRML-ANIL (ours)**            |  68.70 $\pm$ 1.33%  |
> | **FRML-ANIL++ (ours)**          |  **69.14 $\pm$ 1.41%**  |
>
> ***
>
> **Q3**: Minor typos
>
> **A3**: Thank you for pointing them out. We have fixed these typos in our paper.
>
>
> References
>
> [1] Snell, Jake, Kevin Swersky, and Richard S. Zemel. "Prototypical networks for few-shot learning." NeurIPS 2017.
>
> [2] Vinyals, Oriol, Charles Blundell, Timothy Lillicrap, and Daan Wierstra. "Matching networks for one shot learning." NeurIPS 2016.
>
> [3] Kim, Taesup, Jaesik Yoon, Ousmane Dia, Sungwoong Kim, Yoshua Bengio, and Sungjin Ahn. "Bayesian model-agnostic meta-learning." NeurIPS 2018.

---

### Official Review · Reviewer_wsge · 2021-07-17

**Rating:** 7
**Confidence:** 2

**Summary:**

This paper presents a new meta-learning method called FRML. By analyzing functional regions of a base learner, it tries to transfer useful knowledge to a task with insufficient data. Specifically, they present the contrastive learning loss and localization strategy that seems to be the key components of FRML. It is evaluated for two drug discovery tasks, screening and ADMET prediction, which supports their hypothesis well. Several analyses highlight the various aspect of the results.

**Limitations And Societal Impact:**

## Suggestions
* compare with non-meta learning models
* use insightful figures instead of wordy explanation


**Main Review:**

Originality: The originality of this work can be found in the framework and its subunits. They proposed a new loss function and new localization strategy, which are the key ingredient of the new framework.


Quality: They are motivated by the problem of existing models, and hypothesize functionally regionalized knowledge transfer would be useful in low-resource tasks. Accordingly, they proposed FRML, and they validate the hypothesis in the experimental results with two drug discovery benchmarks. However, it's hard to judge if those two benchmarks are low-resource tasks. Therefore, if non-meta learning SOTA models' performance of each benchmark is provided, their validation would be stronger. Just a quick reference of performance, the best score of Tox21 and Toxcast are 0.871 (AUC) and 0.777(AUC), respectively (I understand those two numbers can't be directly compared to FRML's scores).


Clarity: This is generally a well-written paper, however, it'd be better if insightful figures are included instead of an explanation.



Significance: This work deals with one of the fundamental tasks in drug discovery, therefore it would benefit many researchers in this field.

**Time Spent Reviewing:**

6

---

> ### Author Response · Authors · 2021-08-10
> **Response to Reviewer wsge**
>
> Thank you for your review and for acknowledging our contributions and originality. Below, we respond to your concerns point by point. We would really appreciate it if you could let me know whether our response addresses your concerns.
>
> **Q1**: “it's hard to judge if those two benchmarks are low-resource tasks”
>
> **R1**: The two datasets used in this paper are indeed low-resource tasks.
> - In drug activity prediction, almost all assays have less than 100 training samples, and ~50% assays only have less than 50 training samples.
>
> - In ADMET property prediction, as mentioned in Line 313-315 (main paper), we conduct the experiments under the conventional few-shot learning protocol. Each task is a 2-way classification problem with only 5-shot support (training) samples in each class.
>
> ***
>
> **Q2**: Validation of low-resource tasks by comparison with non-meta-learning SOTA models
>
> **R2**: We first would like to clarify that the proposed FRML is a model-agnostic meta-learning method, thus it is ready to boost any non-meta-learning SOTA models as the base learner.
>
> - In our experiments, we use a neural network with two fully connected blocks and a fully connected classifier as the base learner. Each fully connected block contains a fully connected layer, a batch normalization layer, and a LeakyReLU activation layer. Building upon the base learner, we **have compared FRML with non-meta-learning methods (i.e., FC-Individual, FC-All)** and the superiority of FRML over the non-meta-learning methods validates that FRML improves the prediction of low-resource tasks.
>
> - It would be also exciting to investigate more non-meta-learning SOTA models as our base learners. Here, we set the base learner in FRML to be TrimNet [1], which achieves the SOTA performance on Toxcast, and we report its results (denoted as TrimNet-FRML) in Table Re-1. TrimNet-All represents that we directly apply TrimNet to our experiments. We adopt AUC as the evaluation metric. In Table Re-1, we observe that 1) the non-meta-learning SOTA model, i.e., TrimNet-All loses its capacity in our low-resource experimental setup, and 2) TrimNet-FRML effectively improves the predictive ability of non-meta-learning SOTA models.
>
> **Table Re-1**: Comparison between TrimNet-FRML and TrimNet-All. Averaged AUCs with 95% confidence intervals are reported.
>
> |              |       SIDER       |       Tox21       |        MUV        |      Toxcast      |
> |--------------|:-----------------:|:-----------------:|:-----------------:|:-----------------:|
> | TrimNet-All  | 0.767 $\pm$ 0.009 | 0.749 $\pm$ 0.010 | 0.616 $\pm$ 0.017 | 0.768 $\pm$ 0.016 |
> | **TrimNet-FRML (ours)** | **0.791 $\pm$ 0.011** | **0.788 $\pm$ 0.009** | **0.639 $\pm$ 0.014** | **0.795 $\pm$ 0.015** |
>
>
> ***
>
> **Q3**: insightful figures
>
> **R3**: Thank you for your great suggestions. We will add more intuitive figures in the revised paper.
>
> References
>
> [1] Li, Pengyong, Yuquan Li, Chang-Yu Hsieh, Shengyu Zhang, Xianggen Liu, Huanxiang Liu, Sen Song, and Xiaojun Yao. "TrimNet: learning molecular representation from triplet messages for biomedicine." Briefings in Bioinformatics 22, no. 4 (2021): bbaa266.

---

### Official Review · Reviewer_3Ywv · 2021-07-21

**Rating:** 7
**Confidence:** 4

**Summary:**

In this paper, the authors propose a new meta-learning method for drug discovery. The proposed model transfers knowledge from source drug assays to target drug assays. To better leverage the similarity among drug assays, the proposed model disentangles the base learner to different functional regions and reassembles the selected functional regions for each drug assay. The proposed model is evaluated on virtual screening and ADMET prediction tasks and achieves better performance than other baseline methods.

**Limitations And Societal Impact:**

Yes

**Main Review:**

Strengths:

1) This paper focuses on making accurate virtual screening and ADMET predictions with limited drug compounds, which is an important and practical problem in drug discovery.

2) The motivation of the proposed FTML is clear. The authors discuss the similarity among different drug assays and use functional regions to model similarities and differences across drug assays.

3) The experiments are comprehensive. The authors evaluate the performance on both regression (virtual screening) and classification (ADMET prediction) datasets and verify the effectiveness of the proposed method. Besides, the qualitative analysis about learned traces further strengthens the proposed model's motivation and its interpretability.

Weaknesses:

1) It would also be interesting to see a discussion about different layers use a different number of functional regions. From my perspective, lower layers may use less functional regions since they capture more general features than higher layers.

2) In the ablation study III, the authors replace RNN localization network with FC. How do you design the FC network? Do you share the weight across different layers?

Overall, the paper is clearly motivated by comprehensive experiments. I vote for accepting.

**Time Spent Reviewing:**

4

---

> ### Author Response · Authors · 2021-08-10
> **Response to Reviewer 3Ywv**
>
> Thank you for your review and for acknowledging our technical and empirical contributions\. Here, we provide detailed responses to your two questions. Would you please let us know if you have further concerns?
>
> **Q1**: Discussion about different layers using different numbers of functional regions
>
> **A1**: Thank you for your suggestion. We conduct the experiments using a different number of functional regions in each layer. Here, we use two functional blocks in layer one and four functional blocks in layer two. The results are reported in Table Re-1 and Re-2 for drug activity prediction and ADMET property classification, respectively. We observe that the performance is slightly worse than the strategy of using three layers for each layer. One potential reason is that the Morgan fingerprint features have covered enough low-level features, and it might be more useful to add more functional blocks in the first layer.
>
> **Table Re-1**: Performance of FRML with different knowledge blocks in each layer on drug activity prediction. (2, 4) represents that we use 2 blocks in layer 1 and 4 blocks in layer 2. Mean $R^2$ is reported.
>
> |                             | Group I | Group II | Group III | Group IV |
> |-----------------------------|:-------:|:--------:|:---------:|:--------:|
> | FRML-ANIL++ ((2, 4) blocks) |  0.369  |   0.324  |   0.344   |   0.367  |
> | FRML-ANIL++ ((3, 3) blocks) |  **0.375**  |  **0.327**  |   **0.345**   |   **0.372**  |
>
> **Table Re-2**: Performance of FRML with different knowledge blocks in each layer on ADMET property classification. (2, 4) represents that we use 2 blocks in layer 1 and 4 blocks in layer 2.
>
> |                             |       SIDER       |       Tox21       |        MUV        |      ToxCast      |
> |-----------------------------|:-----------------:|:-----------------:|:-----------------:|:-----------------:|
> | FRML-ANIL++ ((2, 4) blocks) | 69.69 $\pm$ 0.85% | **71.28 $\pm$ 0.93%** | 59.72 $\pm$ 1.11% | 73.62 $\pm$ 1.72% |
> | FRML-ANIL++ ((3, 3) blocks) | **70.01 $\pm$ 0.86%** | 71.07 $\pm$ 0.91% | **60.66 $\pm$ 1.09%** | **74.02 $\pm$ 1.57%** |
>
> ***
>
> **Q2**: FC network design in ablation study III
>
> **A2**: In ablation study III, we do not share the weights of the FC network across different layers. Instead, each layer has its own fully connected network since different layers have their specific information.

---

### Decision · Program_Chairs · 2021-09-27

**Decision:**

Accept (Poster)

**Comment:**

This paper seeks to remedy the low-resource drug discovery problem by transferring the knowledge from previous assays, namely in-vivo experiments, by different laboratories and against various target proteins. The authors propose a functional rationalized meta-learning algorithm FRML for such knowledge transfer. The approach appears well motivated and the empirical results appear advance the state-of-the-art.

The reviewers provided detailed reviews providing a long list of pros and a rather short list of cons of the approach/manuscript. They are all coming to the same conclusion that this is a good paper for NeurIPS. I follow the recommendation of the reviewers to accept the manuscript.

From the scores, it may not be in the spotlight or oral presentation range (7), however, I think it could be bumped up if one aims to increase health-related topics in the oral presentation program.